# Participatory Landscape Design and Water Management—A Sustainable Strategy for Renovation of Vernacular Baths and Landscape Protection in Szeklerland, Romania

Albert Fekete *, Ágnes Herczeg, Ning Dong Ge and Máté Sárospataki

Institute of Landscape Architecture, Urban Planning and Garden Art Budapest, Hungarian University of Agriculture and Life Sciences—MATE, Villányi út 35-43, 1118 Budapest, Hungary; herczeg.agnes@uni-mate.hu (Á.H.); ningdongge@phd.uni-mate.hu (N.D.G.); sarospataki.mate@uni-mate.hu (M.S.)
* Correspondence: fekete.albert@uni-mate.hu

**Abstract:** Szeklerland is a historical-ethnic region located on the eastern border of the Carpathian Basin, in the central region of Romania. In Szeklerland, thanks to its varied topography and a network of small settlements, landscape management is still carried out using traditional methods. Szeklerland is a macro-region rich in natural resources. Among its natural treasures, the mineral water springs with healing properties are of particular importance: around 40 percent of Romania's mineral water resources are found here. This richness in hydrogeological features is due to the fact that the post-volcanic activities in the young tertiary mountain ranges in the region still produce large quantities of carbon dioxide, which dissolves beneficial minerals from the earth. When dissolved in water, these minerals produce mineral waters that can be used to cure various types of diseases. For centuries, the medicinal properties of the mineral waters of Szeklerland have been regularly used by the local population. In addition to their consumption, small and larger vernacular baths were built in the settlements with medicinal springs, and their regular use led to the development of a traditional, local cold-water bathing culture in the region. However, the vernacular baths were destroyed in the world wars, and their traditional use was abolished by the apparatus of the 20th century communist regime, which had no respect to natural and cultural heritage. After the political change in 1989, the attention of the society turned back to tradition and values. Alongside (or as part of) nature and landscape conservation initiatives, the reinterpretation and restoration of the intangible and practical values of vernacular baths in Szeklerland also began. Over the past decades, the renovation of vernacular baths, which started as a professional–civic initiative, has grown into an independent heritage conservation programme: dozens of vernacular baths have been renovated in Szeklerland over the past twenty years with public participation initiated and led by professionals. In the course of the renovations, baths used by local communities have been rebuilt using nature- and environment-friendly techniques, materials and in a way that they are also related to the physical environment and the mythology of the region. The project has won prestigious awards both in Romania and internationally, and has become a successful and exemplary movement in landscape heritage conservation.

**Keywords:** cultural heritage; traditional land use; bathing culture; community building

## 1. Introduction

The nature-based land use in Szeklerland is the result of the traditional coexistence of man and nature, and the close relationship between communities and the landscape. The complex geofactors have created a distinctive and unique landscape potential in the region. Due to the isolated geographical location of the area, it has been mostly peripheral to economic development and innovation over the centuries, and modernisation processes have been delayed. Hence the forms of land use resulting from this isolation are

rational, based almost exclusively on internal resources that are renewable and sustainable. The archaic Szekler landscape has become a model for the conservation of natural and cultural heritage. Natural resources and traditional land use remain the basis for landscape development strategies in the region, with the varied and multipurpose use of medicinal mineral springs as a key pillar [1–3].

The use of mineral waters for various treatments is probably as old as humanity. Hydrotherapy is one of the basic methods of treatment widely used in natural medicine, also called as water therapy, aquatic therapy, pool therapy, and balneotherapy. Use of water in various forms and of various temperatures can produce different effects on different systems of the body [4–7]. The strategic renovation of the vernacular baths in Szeklerland will help to continue and develop this tradition.

### 1.1. The Landscape

The area of Szeklerland is 12,450 km$^2$, its geographical borders are shown in Figure 1. In terms of its topography, it consists of a varied mixture of lowland, upland and highland landscapes, where the mountainous structural and relief elements have a prominent role [8].

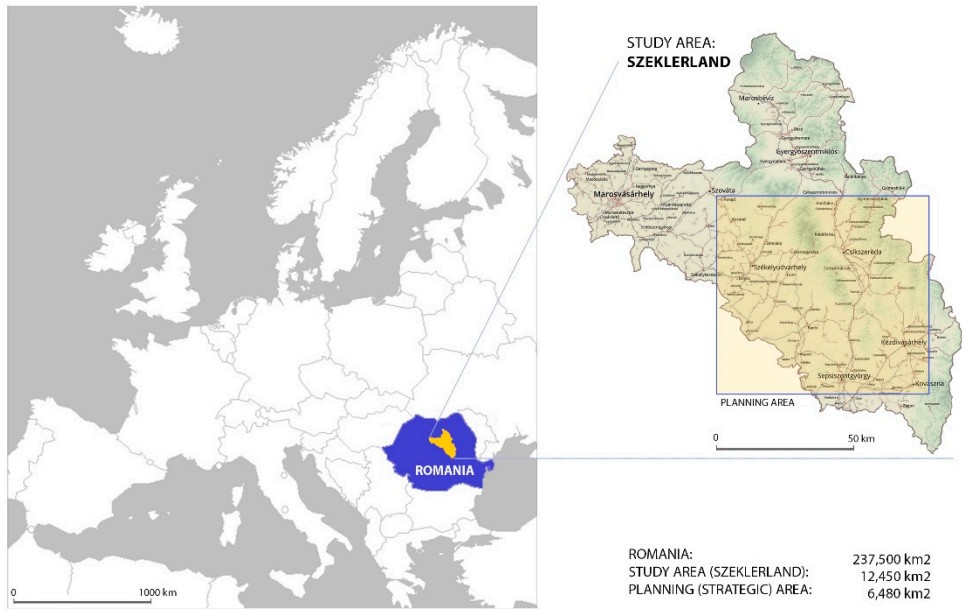

**Figure 1.** Location of the study area (Szeklerland) and planning (strategic) area on the map of Europe. Source: prepared by the authors based on the map of Szeklerland [9].

The varied landscape is the result of less than 40,000 years of volcanic activity, which has created a highly fragmented surface morphology, which is difficult to navigate [10,11]. Its varied geology, picturesque rock formations, karstic features, rich network of rivers and streams, extensive forests and subalpine regions are characteristic components of its picturesque landscape. No wonder that from the 18th century there have been a stream of descriptions and travelogues depicting the natural beauty of the landscape, from both the figures of Hungarian literature and sciences [12–21] and several Western European travellers [22–25].

Based on the complex spatial and aesthetic experience in landscape perception, Szeklerland meets the definition of cultural heritage of the European Landscape Convention [26]. It has preserved its natural treasures and features and also bears the marks of the man who cultivated the landscape, with the spiritual legacy of the community that lived in it.

Centuries of traditional farming have created a characteristic landscape and settlement pattern in Szeklerland. The topography, hydrography, climate and biogeography of the area have created the special spatial position and internal structure of the settlements and the farming systems (in the mountains mainly livestock farming, forestry, terraced

farming, traditional property patterns, allotments, hedgerows). More than two thirds of the settlements are located in the spring areas, along river valleys and on valley terraces.

Larger villages with extensive agricultural land have developed on the edges of the basins, while small villages and hamlets are located along smaller rivers and streams [27].

The villages were based on the "tens", which, in accordance with the military way of life, represented the smallest unit of the army and the related families into a highly rational form of settlement, which can still be detected in mental mapping today [28].

This specific landscape pattern forms the basis of a heterogeneous but well-managed landscape heritage, which is partly the result of agriculture and partly of mountain livestock farming. Linked to the latter is the system of dual place of living, a practice which has now disappeared from European land use. In keeping with the rhythm of the year, the animals are driven out to the mountain pastures on St. George's Day (24 April) and then back to the village stables on St. Michael's Day (29 September) at the beginning of autumn.

With the development of the "tens" and the increase in population, more and more remote and steeper areas had to be farmed. The new pastures were created by clearing the forest, and in the areas ploughed perpendicular to the slope, formed by centuries of ploughing, agricultural terraces without retaining walls had been gradually developed [29–33]. With the emigration of the population (already in the 19th century) and the decline of agriculture, the agricultural terraces were once again used as hayfields, which nevertheless retained their terraced character, giving the local character of the hillsides.

Regarding landscape aesthetics, the cultivated mosaic parcels of land are also dominant features [34–37]. In the (periodically redistributed) allotment system of land, which originated from community land ownership, it was inconceivable that vast monocultural plantations could be established. Individual families cultivated their land according to their own needs and ideas, creating a diverse, mosaic landscape, the aesthetic appearance of which were further enhanced by tree planting along the edges (Figure 2).

This local character of the landscape has survived the destructive effects of forty-five years of Soviet rule on landscape and society (large-scale industrial agriculture, collectivisation, nationalised forestry, etc.) and the arbitrary, chaotic changes in ownership and land use after the change of the regime in 1989, and is ready for renewal and further creation of value through the recent social self-organisation and professional activity. The traditional property structure still makes it difficult to make farming competitive [38,39].

Historically, the basic activities and main economic resources of the highland landscape have been forestry and wood processing. After the end of communism, the institution of public ownership of forests and mountain pastures was restored in Szeklerland. Forests are in undivided community ownership, where owners share both the duty of cultivation and the benefits. Wood processing, traditional folk woodworking, carpentry and joinery have become basic skills in this area [40,41].

### 1.2. The Significance of Mineral Waters from Medical and Landscape Aspects

The diversity of the landscape in Szeklerland is due, among other things, to the natural features resulting from its geological characteristics [42,43]. One of the main groups of these are the medicinal mineral springs and mofettes, which are the subject of this study and were the basis of a significant bathing culture developed over the centuries (Figure 3).

According to the Directive 2009/54/EC of the European Parliament and the Commission, mineral water is defined as a "microbiologically wholesome water originating in a groundwater table or deposit and emerging from a spring tapped at one or more natural bore exits. Natural mineral water can be clearly distinguished from ordinary drinking water: (a) by its nature, which is characterized by its mineral content, trace elements or other constituents and, where appropriate, by certain effects; (b) by its original purity. The characteristics referred to in point (a), which may give natural water properties favourable to health, shall have been assessed from the following point of view: geological and hydrological, physical, chemical and physico-chemical, microbiological, if necessary, pharmacological and clinical".

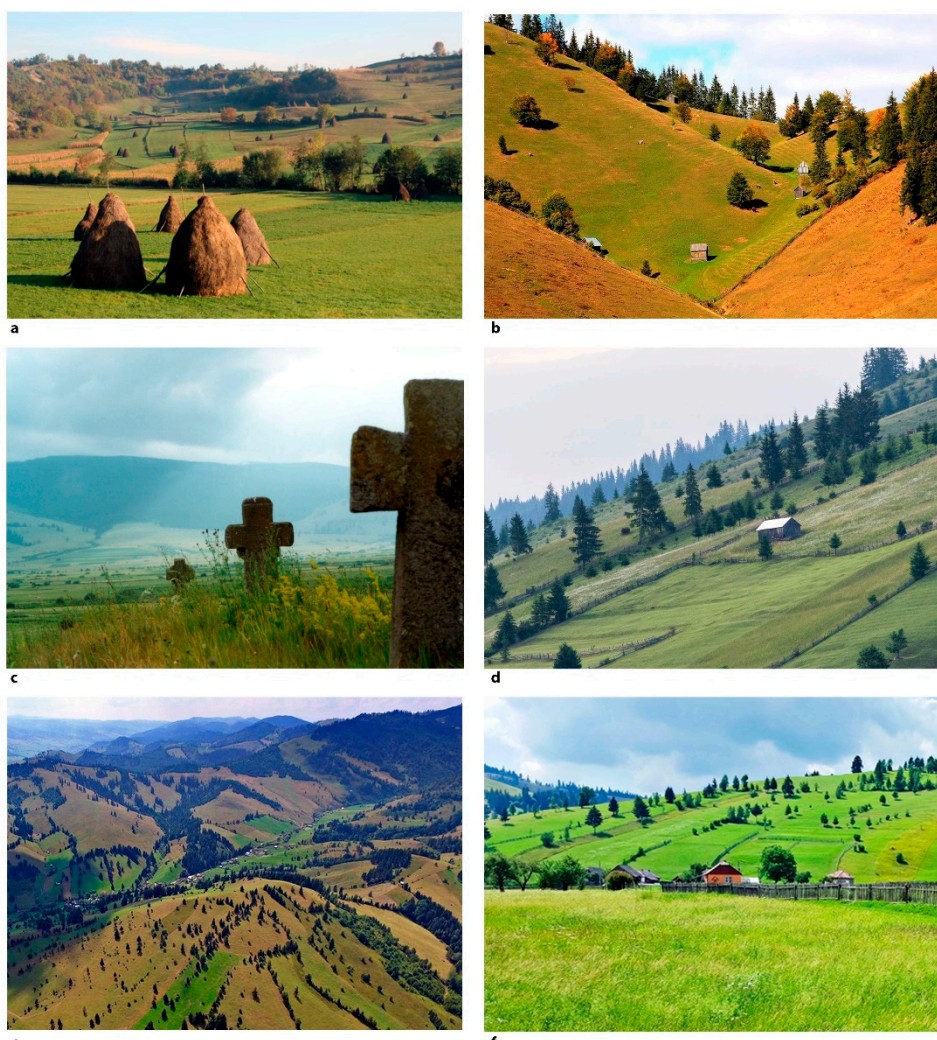

**Figure 2.** Landscape and land use in Szeklerland. Source: photos by the authors (**a**,**b**,**e**,**f**) and by Ars Topia Foundation (**c**,**d**). The figures show the mosaic-like landscape shaped by traditional use of parcels (grassland, meadow) and the land occupation typical for mountain villages.

Three types of mineral waters were defined according to the total dissolved mineral content: very low mineral content, with mineral salt content, calculated as a fix residue, under 50 mg/L, low mineral content, not greater than 500 mg/L; and rich in mineral salts, value greater than 1500 mg/L [44].

The concentration of mineral waters in Szeklerland varies between 500 mg/L and 200,000 mg/L with varied content of carbon dioxide or other gas, and temperatures between approximately 6 °C and 40 °C [46]. In particular, on the basis of their temperature, the mineral waters of the sites analysed in this study can be classified as cold thermal waters (below 31 °C). Even in a relatively small area, the composition of mineral waters is extremely diverse. Particularly common are mineral waters with high salt, sulphur or iron content, which have traditionally been used for thousands of years in two main forms, drinking water and bathing water, according to their mineral content (Figure 4).

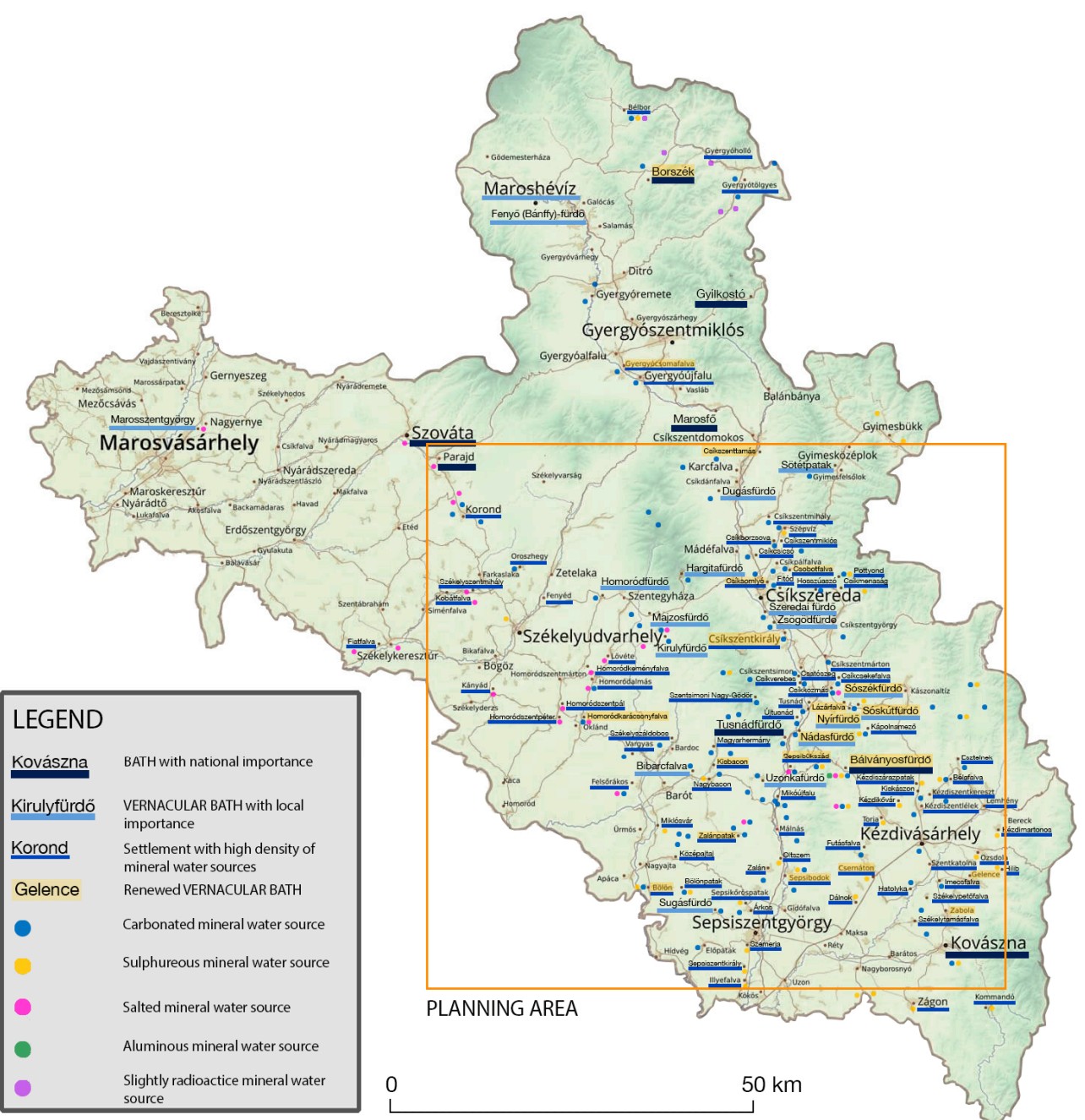

**Figure 3.** Map of mineral water sources of Szeklerland. The map shows among different hydrogeological aspects the location of bath building volunteer camps in Szeklerland. Source: prepared by the authors and technical contribution of Maja Erdei based on the map of Zsigmond Enikő [45] and own research of the authors.

The water from the mineral springs of Szeklerland began to be bottled in the 18th century, first in earthenware jugs and then in glass bottles. According to documents, the first bottled water in Szeklerland was 'Korond water' in 1792, followed by 'Borszék mineral water', which was called the king of mineral waters, in 1806 [47–52].

Around the numerous (hundreds of) mineral springs, smaller and larger baths were developed over time, which, according to a study by the surgeon and obstetrician Károly Cseh from 1873, helped cure various illnesses, both in the form of drinking and bath cures, such as stomach and intestinal disorders, anaemia, disorders of the excretory and

respiratory organs, treatment of female ailments and male "inabilities", musculoskeletal disorders, cardiovascular complaints, nervous weaknesses [53].

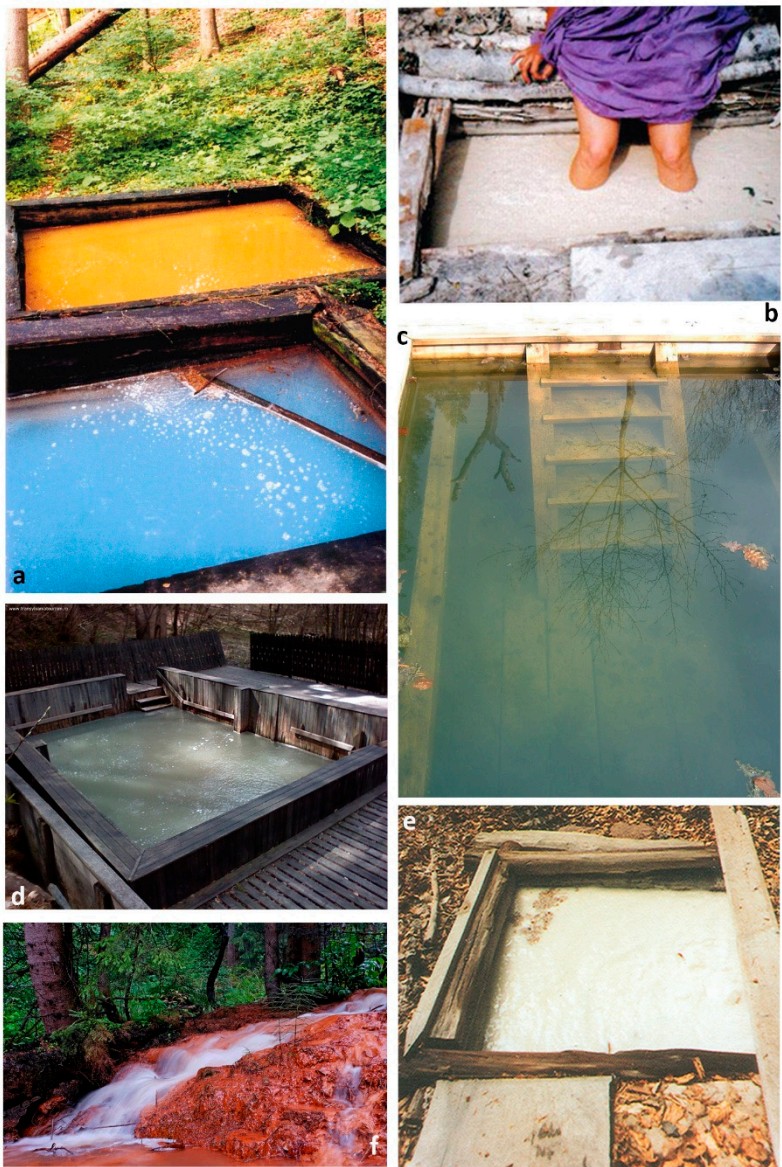

**Figure 4.** The colours of mineral waters differs depending on their various chemical composition: as far as the high concentration of iron induce a red or yellow coloration (**a** (**above**),**c**,**f**), the blue (**a** (**below**)) and grey (**b**,**d**,**e**) tones show the high sulphurosus content of the water. Source: photos by Ars Topia Foundation.

Descriptions from the 18th and 19th centuries, in which local residents report on the beneficial and healing effects of regular daily bathing, and archival photos demonstrate the lively bathing life in Szeklerland [54,55], (Figure 5).

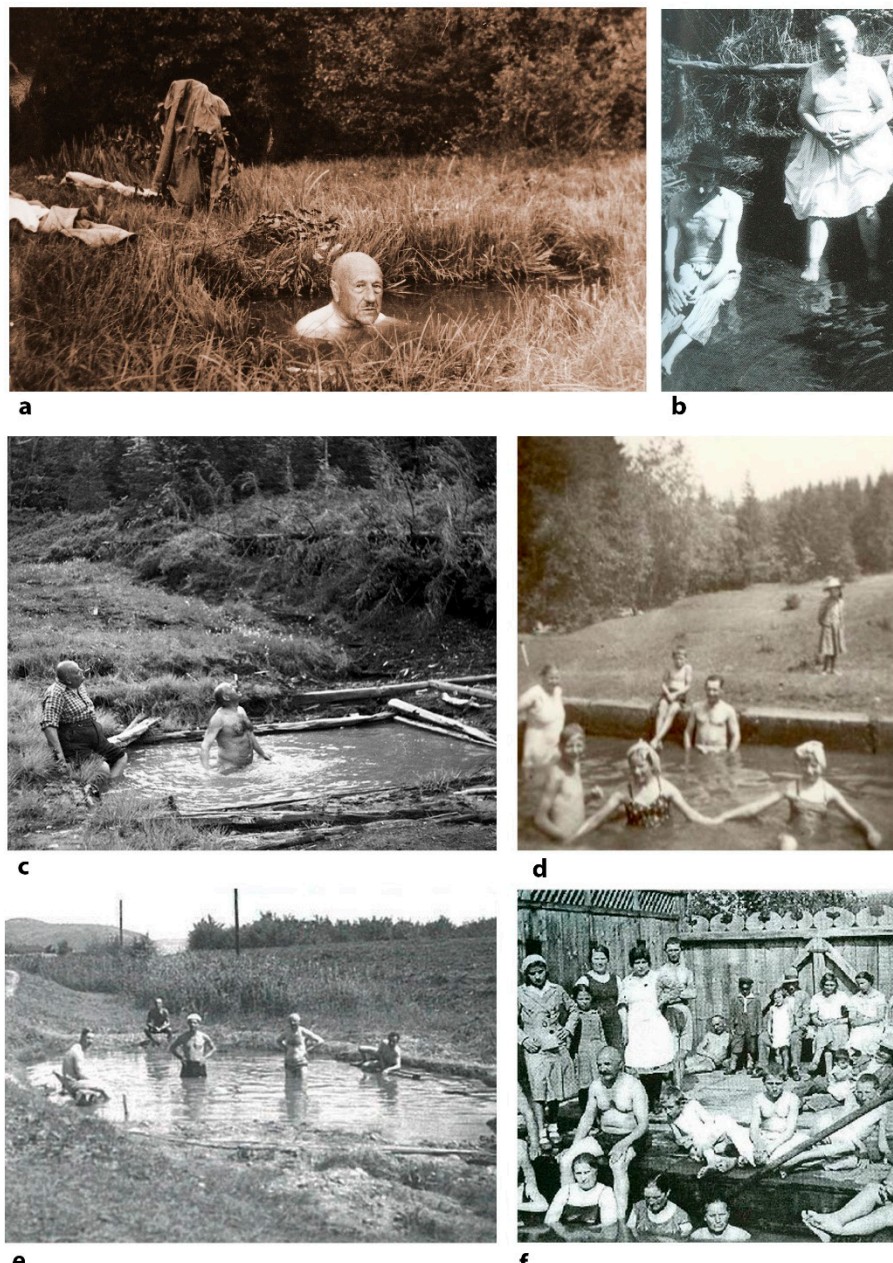

**Figure 5.** Use of traditional mineral baths by local people at the end of 19th century and first part of the 20th century: (**a**,**b**)—Nádasfürdő (Source: Ars Topia Foundation); (**c**)—Torja (source: photo collection of Szkler National Museum Sepsiszentgyörgy); (**d**)—Kirulyfürdő (source: private collection of Csaba Jánosi); (**e**)—Sósfürdő, Székelyudvarhely (source: Vofkori György [56]); (**f**)—Homoródfürdő (source: private photo collection of Márton Balázs).

During their history, there are two trends in the use of baths. The majority of baths have remained local, linked to a village or settlement. These vernacular baths, which preserved the natural state of the surroundings with minimal infrastructure (wooden pools, ladders), were mainly used by local people [57,58].

A small number of baths, thanks to their waters richer in minerals and favourable geographical location, developed into spa towns from the 1870s, provided with all the services and facilities needed for modern spa tourism. Drinking and bathing saloons, outdoor and indoor pools, sitting baths, mofettes, fountains, hotels, restaurants, theatres, parks, summer gardens, forest walks, pavilions and lookouts, etc., with architecture and

facilities typical of the civic culture of the second half of the 18th century, were built, giving these spa towns a distinctive urban character and landscape [59–61].

However, the unfavorable political developments of the 20th century history destroyed the spas and almost permanently destroyed the vernacular baths and the traditions of their use, which made up the majority of this colourful bathing culture. The nationalisation that followed the Second World War put a brake on dynamic development ideas of local baths, and almost completely eradicated the traditional and civic bathing cultures. The state did not support the development of medical springs, claiming that they could dry up and that the public work of maintaining public baths had lost its value and credibility. For forty years from the 1950s, rural life and the traditional way of life in Szeklerland, including the folk bathing culture, were characterised by a period of stagnation and decay [62–64].

As a consequence, by the 1990s, a deplorable state of neglect had developed: the mineral water springs were clogged, the pools silted up, reclaimed by forest and vegetation, and the buildings had disappeared. In less than four decades, the results of centuries of conscious human activity, conservation and nature-friendly development have been destroyed (Figure 6).

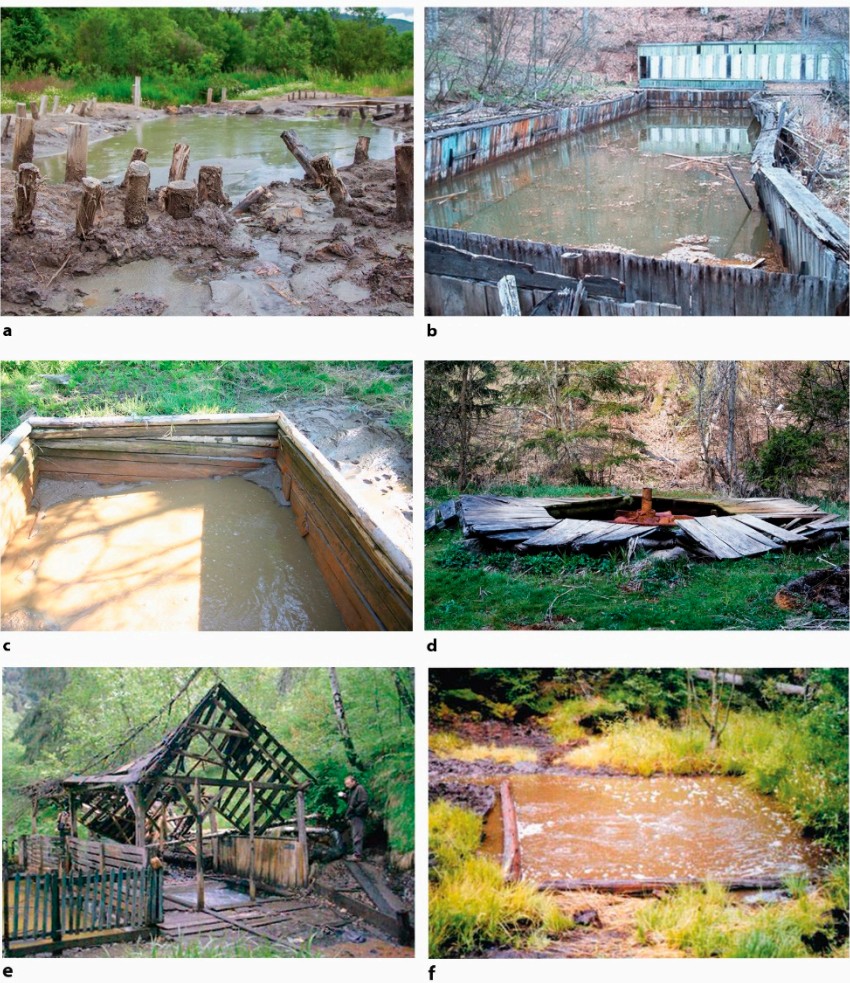

**Figure 6.** Ruined traditional baths in the 1990s: (**a**)—Parajdfürdő, (**b**,**e**)—Csiszárfürdő, (**c**)—Hammasfürdő, (**d**)—Szelterszfürdő, (**f**)—Vallató (source: Ars Topia Foundation).

## 1.3. Health and Landscape Aspects of the Bathing Culture in an International Context

The therapeutic use of the mineral waters—and especially of the hot springs—has been prevalent in Europe from ancient times to the present day. Baden-Baden, Bad Ischl, Bath, Budapest, Karlovy Vary, Spa or Vichy are only a few of the most famous European spa towns, but Europe is home to many more thermal spa settlements with unique urban

character, different styles of architecture, and different spa traditions, built around bathing in or drinking of the thermal waters. This spa culture, in all its variety and different local flavours, can truly be considered a unique European heritage. The spa culture played a prominent role in the development of landscape architecture and open space design, influencing in a positive way the perception and environmental culture of the citizens; the building, use and maintenance of spas highlighted the importance of water and other natural values [65–67].

Thermal towns were the "cafés of Europe", places where members of all strata of society could mix and exchange ideas—where the rules ensured civilised conduct. Thus, spas have played a leading role fostering peace, co-operation and creativity, protecting the built and natural environment, and promoting sustainable cultural development, a role that has been present throughout European history and continues to this day [68–70].

The importance of the European spa culture is proved among others by the fact, that the "European Route of Historic Thermal Towns" is one of the cultural routes of the Council of Europe, certified in 2010 [71].

In Europe, tourism and health services are almost exclusively based on the therapeutic and recreational use of thermal waters, whose positive physiological and recreational effects are well known. The Szekler cold bathing culture examined in this study differs fundamentally from the culture of thermal baths. The cold mineral waters of the Szekler vernacular baths have been proven to have general corroborative, powerful immunostimulating, antidepressant, stress-relieving and other medicinal effects, and there is extensive international medical and balneological literature on the medicinal effects of cold-water baths and mineral waters [72–80].

The regular use of medical baths by local residents, mostly in the summer season, is a part of the relaxation and cleansing after working in the fields/forests. Over the centuries, it has become part of daily routine as a natural remedy that stimulates circulation and metabolism, regenerates muscles and reduces arthritic pain, and its curative effects have contributed to the development of respect for nature and the need to preserve and maintain the natural environment over generations.

The conscious choice to stay relaxed in an uncomfortable situation (a cold mineral water basin) is itself a powerful mental exercise that builds self-confidence and resilience, which can have flow-on effects on the rest of your life.

Despite their recognised medical effects, the health and recreational use of cold-water baths has not spread widely. The example of Szeklerland is completely unique in Europe, because although cold water bathing has a centuries-old tradition in several Baltic and Scandinavian countries (cold water dipping after sauna bathing, medical or ritual ice bathing), none of these bathing customs is based on mineral waters with medicinal properties.

In the course of our research, we have not found any references in the relevant literature (medical, tourism, ethnographic, hydrogeological, landscape architecture, etc.) which would suggest a European practice similar to the Szekler folk mineral bathing culture. The literature and internet search has been extended to other continents.

Based on our results, we can say that in terms of the content of mineral waters, we have not found a hydrotherapeutic bathing culture similar to that of Szeklerland. In the places found after international research (Japan, China, Taiwan), not mineral water, but seawater or fresh water without significant mineral concentration is used for cold water bathing [81].

There are, however, many more similarities and parallels between the vernacular baths of Szeklerland and the internationally known examples and sites in terms of size, scale, location, landscape setting, natural environment and lower intensity of use. In this context, it can be noted that this tradition of bathing based on local use is typical for small communities all over the world, who use baths occasionally for short periods but on a regular basis. Bathing pools, created and maintained using traditional techniques, simple tools, natural or nature-friendly materials, with small-scale landscaping (at the level of

object design) and community work, are everywhere integrated into the landscape in terms of their design, ecological and sustainability characteristics [82].

## 2. Materials and Methods

This paper discusses the landscape aspects of the unique vernacular baths based on mineral water in Szeklerland, and the process of the renewal of the environment of the baths. The article emphasizes the professional background and social significance of the bath renovation strategies.

Over the last twenty years, several folk baths in Szeklerland have been renovated with the involvement of the local community and representatives of various disciplines (landscape architecture, hydrogeology, architecture, social sciences, etc.). The work has been initiated and managed from the very beginning by the Ars Topia Foundation [83] and its professional partners.

The renewal of the vernacular baths in Szeklerland represents a social demand-led, sustainable environmental renewal programme realised with social participation, of which similar architectural initiatives are known worldwide [84–87]. According to the research by Csaba Jakab, the model of the bath building volunteer camps in the Carpathian Basin with a creative-educational purpose can be traced back to the 1970s, to the architecture camps of the BME in Visegrad, followed by several other initiatives [88,89]. The 2014 Venice International Architecture Biennale presented a number of volunteer building camps, among which vernacular bath buildings were represented in significant numbers [90]. The first collaborative building project of the Ars Topia Foundation was realized in 1997, specifically for landscape architects, in order to preserve and develop the landscape heritage in the Dörögd Basin, Hungary [91].

The number of mineral water springs and vernacular baths in Szeklerland is not known exactly, but it is estimated at several hundred. The harmony they represent between man and nature, their functional design, and traditional landscape imprint, make them characteristic sustainable elements of the Szeklerland landscape [92–96].

### 2.1. The Rationale behind the Bath Renovation Strategy

The traditional use of the destroyed mineral springs and folk baths is still alive in the memory of village communities. From the very beginning, the maintenance and environmental enhancement of vernacular baths of local significance was the responsibility of village communities, so it was clear that, in the spirit of sustainability and community development, successful renewal and future maintenance should involve local communities.

Accordingly, a participatory planning and building process has been visioned and realised in case of vernacular bath renewals as well. In the case of the pilot project, launched in 2000 in Lázárfalva (Lazaresti, Romania), three main pillars of the long-term renewal strategy were defined, which served later on the mission of the renewal strategy:

- Community collaboration.

In the early 1990s, without financial resources, the local communities were in amnesia of losing their traditions. Considering these social and economic conditions, the most effective way to renovate folk baths was through the traditional form of community work, which is typical of Szeklerland. The main idea of the volunteer building camps is that everyone contributes to the community work what he or she has strength and knowledge in, and can afford: wood, transport, tools, food, accommodation, money, manual labour, skills, etc. The key to the success of the volunteer camps, launched in the early 2000s by the Ars Topia Foundation, is the innovative way in which the form of organisation that exists in the collective memory of local communities, cooperation based on solidarity and brought to life by real need, has been revived, opening it up to participants from other parts of the Carpathian Basin. In this way, besides municipalities, craftsmen and local helpers, university students, landscape architects, architects and geologists, professional and non-governmental organisations and churches were actively involved in the organisation, construction and implementation, leading to a broad dissemination of the method.

- Economic and ecological sustainability.

The renewal and reconstruction of vernacular baths are at a level that can be accomplished with local, internal resources and a little external assistance, using the traditional working method described above. (The renewal of larger, urban spas is a difficult task, due to the confusion of ownership, the difficulties of privatisation processes and the greater demand for funds). The renewal of vernacular baths is characterised by sensitive interventions that are adapted to the site and respectful of nature and the environment. In addition to aesthetics, this approach also promotes sustainability and biodiversity. Apart from the natural features and land cultivation, the sites incorporate a number of specific architectural and cultural monuments, heritage and traditions, thereby they fulfil the quality requirements of the landscape heritage of Szeklerland. In the context of the use of natural resources, the use and intervention of vernacular baths typically only reach a degree that is strictly necessary to satisfy the bathing and the related functions [97]. No disproportionate paved surfaces, inappropriate materials, oversized infrastructure.

- Environmental education.

One of the aims of the bath renovation strategy is that during the collaborative planning and construction work, the participants from the village and from other places (including abroad) will learn from each other how to recreate landscape heritage elements, how to preserve and use natural values in a sustainable and civilised way. The aim is to revive methods of natural erosion control, local traditional, often forgotten, building techniques, traditional Szekler ornamental painting, and to acquire knowledge of local materials and traditional craftsmanship in shaping wood and stone. In order to raise awareness of all this, invited guest speakers give lectures in the evenings during the camps on the topics of landscape, nature and heritage conservation, rural and regional development, ethnobotany, ethnography, protection of architectural features and the traditional character of villages.

Beside the mineral water, the post-volcanic activities also result in gas escapes, the medicinal properties of which are used in so-called mofettes (gas baths). The minerals dissolved in the gas, as well as the carbon dioxide, are excellent for curing serious vascular diseases. Mofettes, which were built in addition to vernacular baths, also proved to be popular and later became part of the civic bathing culture. Mofettes were also rebuilt in several places during the volunteer building camps.

### 2.2. The Scenario of the Bath Renovations

The renewal of the vernacular baths involves several dozen participants per site. In each case, the event itself takes around 10 to 14 days, but it is preceded by a much longer preparatory process lasting several weeks. Although each location and each community have its own special characteristics, there are some general principles and criteria that can be followed. According to the two decades of experience, the most effective scenario for organising and managing the event can be summarised in the following points [98].

#### 2.2.1. Selection of the Location for the Bath Building Camp

- It is selected from hundreds of mineral water springs of Szeklerland at least a year before the camp, allowing enough time to prepare.
- The mineral water spring (the bath) should be located in public space that is always open to everyone.
- If a request is received from a local community, this could be a priority criterion for selection.

#### 2.2.2. Preparation for the Camp

- *Identifying team leaders.* A team leader can be someone who has already participated in a camp and has the desire to work as a team leader. The team leaders decide among themselves who will take on which task with their group.

- *On-site visit*. The purpose of the field trip is to become acquainted with the site and its history, to make a survey (drawings of the site, photo documentation), to meet the local community and to understand their needs.
- *Landscape design*. Based on a concept developed collaboratively by the preparatory team and the local community, each group prepares the design on a given subject (e.g., soil erosion control, habitat protection) or object (wellhead installation, seating pool, swimming pool, changing rooms, toilets, pavilion, stairs, boardwalks etc.). This work involves landscape architects and architects or students of these disciplines (e.g., in the context of a university assignment or a student design competition). The plans will be presented to the local residents in a village forum, where they will have the opportunity to comment, discuss, and develop the idea (Figures 7–9).
- *Village forum*. Organised by the local community. Organised by the local community, its purpose is to enable the participants of the building camp and the local residents to become to know each other and to plan together in advance.
- *Cooperation agreement*. Established between those coordinating and organising the camp, it includes commitments and obligations (not only for construction but also for maintenance afterwards).
- *The raising of financial resources*. The work of all participants is a voluntary donation. Most of the volunteers are university students. The organisers will also obtain grants to run the event (e.g., for the transport of volunteers). The local people organise the local activities (accommodation and food for the participants, purchase and transport of building materials, supply of tools, presence of local craftsmen, organisation and implementation of fundraising event in the village for the camp).
- *Announcement of the camp*. The building camps are organised as an open volunteer movement. The main means of recruiting participants is through university announcement, but those who have already participated spread the message among their friends. It can be considered as an internship for university students.

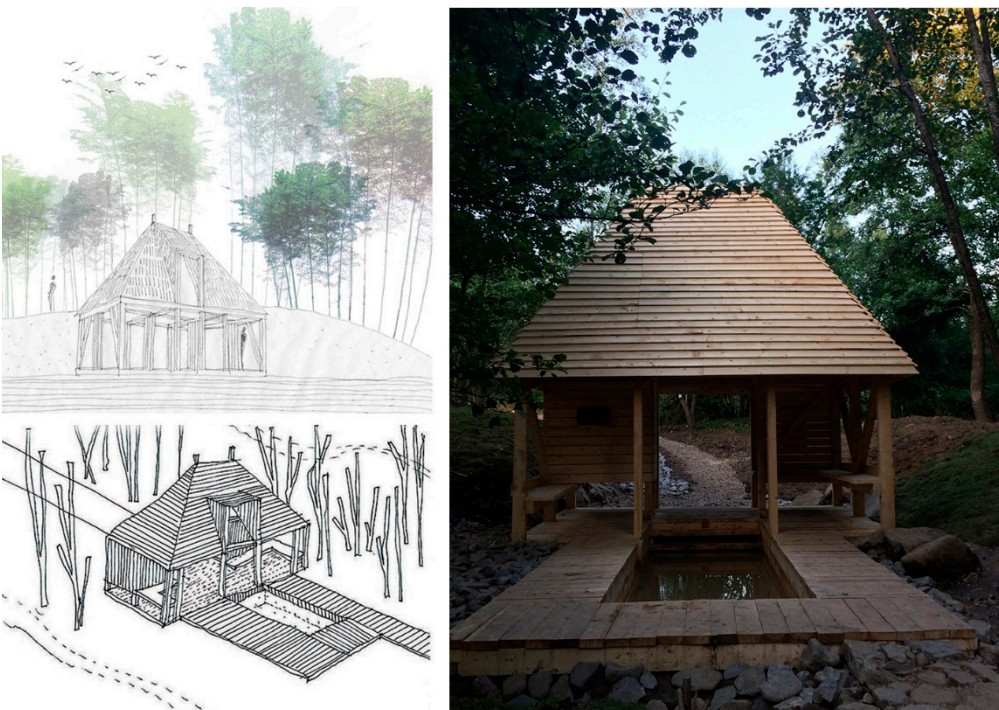

**Figure 7.** Concept and renewed site of the Csipkés mineral water bath in Zabola (Zabala, Romania), as result of the sustainable strategy for renovation of vernacular baths in 2018. List of designers and contributors on the https://www.arstopia.hu/2018-kalakak-zabola (accessed on 5 December 2021).

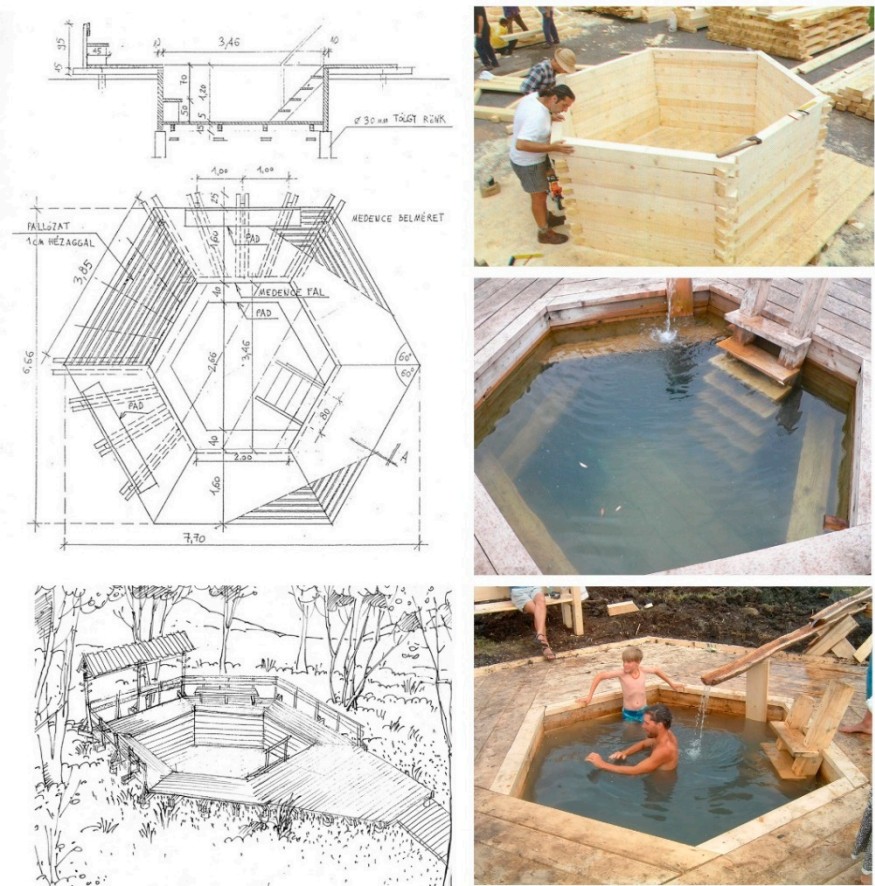

**Figure 8.** Design details, concept and renewed basin of the Szent Anna bath in Csíkszenttamás (Tomesti, Romania) in 2007. List of designers and contributors on the https://www.arstopia.hu/2007 kalaka (accessed on 5 December 2021).

2.2.3. Managing the Building Camp

- *Launching the camp*. Some of the organisers travel to the site a few days before the start, organise the arrival, or, if the terrain is difficult and the work is heavy, carry out professional preparatory work (mowing, shrub clearance, marking out objects, etc.)
- *Community work*. Once the participants have arrived, the construction work starts with professional supervision. The camp is not only about building, but is also a creative event, with a rich programme of community-building and intellectual activities. (Figure 10).
- *Ceremonial opening*. On the last day of the camp, the renovated bath is officially opened and inaugurated by a common barbecue gathering and celebration of the participants and the entire village community.
- *Afterlife of the renewed baths*. Ensuring that the objective is achieved: the attachment to the place created by collaborative work also helps the local community to care for and maintain the renovated baths in the long term through their own efforts.

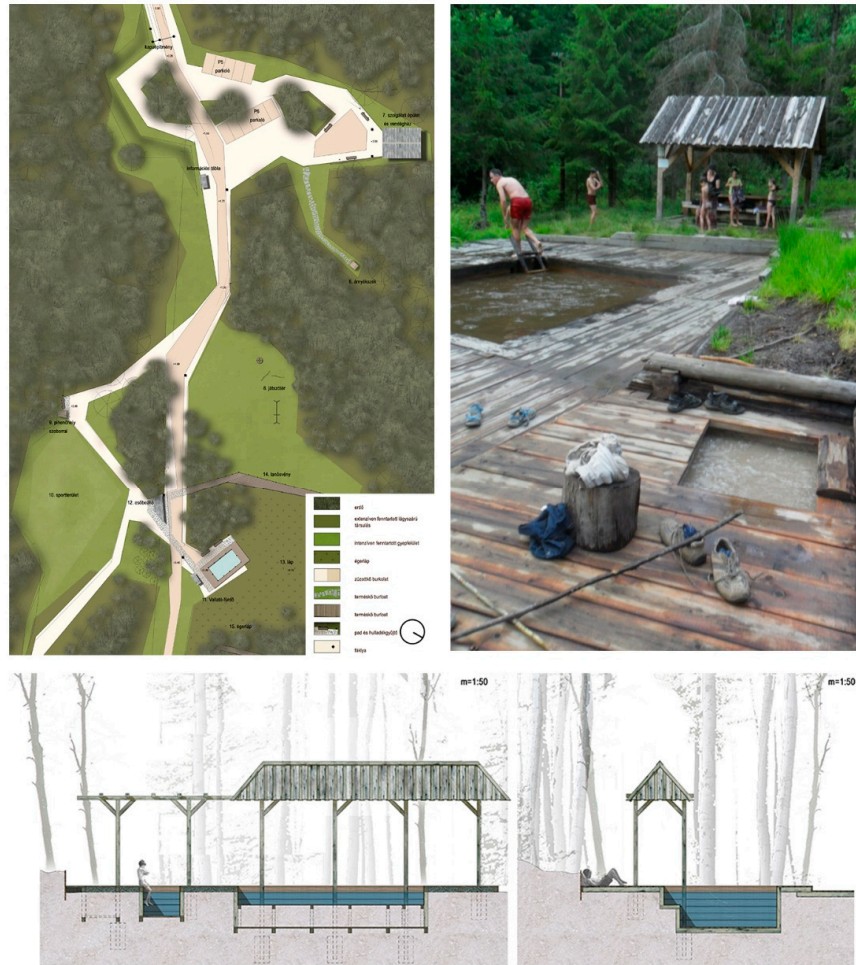

**Figure 9.** Design details and photo of the renewed site of Hammas bath in Sepsibükszád (Bixad, Romania), 2010. Design by Máté Sárospataki and János Hómann.

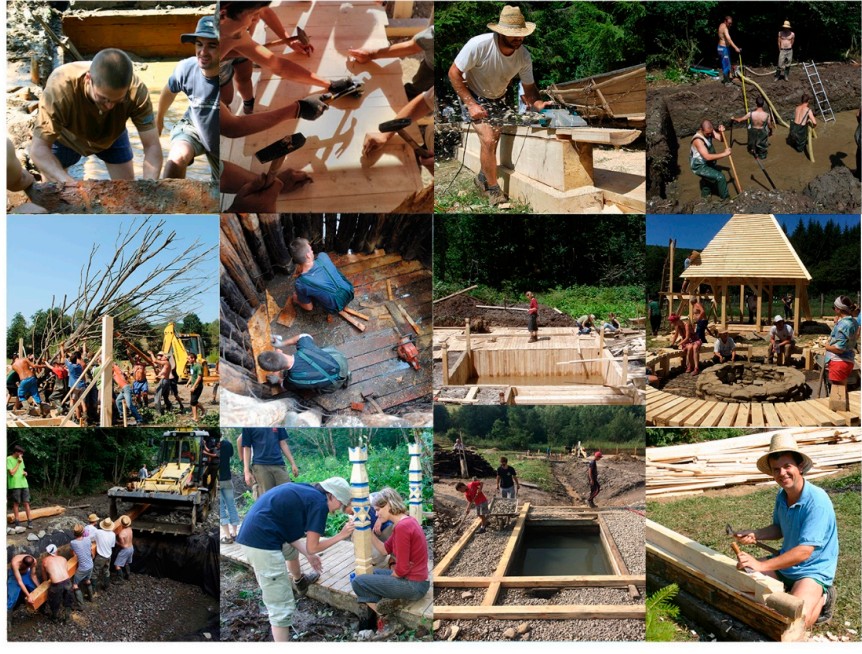

**Figure 10.** Instant photos of community work. Source: Albert Fekete, Gábor Kulich, Máté Sárospataki, Ars Topia Foundation.

### 3. Results

*3.1. Renovation of Twenty-Two Folk Baths in Szeklerland*

In the last twenty years, twenty-two folk baths have been renovated during the bath building volunteer camps organised in Szeklerland, which started in the early 2000s. The geographical location of the renewed sites within the region is illustrated in Figure 3. Table 1 lists the sites in chronological order of renovation.

**Table 1.** List of the bath-building volunteer camps in Szeklerland, in chronological order. The table shows certain sites repeated. Several rehabilitation landscape interventions for restoration purposes have been carried out on these sites over a number of years.

| No | Year of Community Work | Name of Bath | Hungarian and Romanian Name of the Settlement |
|----|------------------------|--------------|------------------------------------------------|
| 1 | 2001 | Nyírfürdő | Lázárfalva/Lazaresti |
| 2 | 2002 | Nádasfürdő | Tusnád/Tusnad |
| 3 | 2003 | Sószékfürdő | Csíkkozmás/Cozmeni |
| 4 | 2004 | Sóskútfürdő | Kászonújfalu/Casinu Nou |
| 5 | 2005 | Borsáros fürdő | Csíkszentkirály/Sancraieni |
| 6 | 2006 | Kerekeger feredő | Csobotfalva/Cioboteni |
| 7 | 2006 | Barátok feredője | Csíksomlyó/Sumuleu-Ciuc |
| 8 | 2006 | Dungó feredő | Homoródkarácsonyfalva/Craciunel |
| 9 | 2006 | Tókerti tanoda | Lázárfalva/Lazaresti |
| 10 | 2007 | Szent Anna feredő | Csíkszenttamás/Tomesti |
| 11 | 2007 | Vallató fürdő | Sepsibükkszád/Bixad |
| 12 | 2008 | Csomafalvi feredő | Gyergyócsomafalva/Ciumani |
| 13 | 2009 | Tündérkerti feredő | Borszék/Borsec |
| 14 | 2010 | Lázárfalvi kút | Lázárfalva/Lazaresti |
| 15 | 2010 | Hammas fürdő | Sepsibükkszád/Bixad |
| 16 | 2011 | Süttei feredő | Sepsibodok/Bodoc |
| 17 | 2012 | Malom és Csókás feredő | Csernáton/Cernat |
| 18 | 2013 | Maczkó bácsi feredője | Kisbacon/Batanu Mic |
| 19 | 2014 | Bugyogó feredő | Zalánpatak/Valea Zalanului |
| 20 | 2015 | Apor lányok feredője | Bálványos/Balvanyos |
| 21 | 2016 | Büdös feredő | Bölön/Belin |
| 22 | 2017 | Nádika fürdő | Gelence/Ghelinta |
| 23 | 2018 | Csipkés feredő | Zabola/Zabala |
| 24 | 2019 | Dungó feredő | Homoródkarácsonyfalva/Craciunel |

The bath renovations are a living example of tradition, natural medicine, environmental protection and community building for the local community. They raise awareness in local communities and the wider society of the importance of water, its medicinal properties and traditional ways of drinking and bathing.

*3.2. Broad Professional and Social Participation*

In addition to the local residents, a significant number of university students, mainly of landscape architecture and architecture, had also participated the camps. The camps were organised, run and supported in cooperation of government agencies, NGOs, professional organisations, entrepreneurs, churches and educational institutions, as a spontaneous, grassroots initiative. Over the past 21 years, more than 10,000 people have participated in the camps, including volunteers, local residents and supporters, which shows the social importance of the programme.

The work is useful for heritage protection and landscape conservation, but its greatest benefit is the awakening of the community to the fact that there are internal resources in the landscape, in the community, and that all it takes is will, cooperation and trust to build a livelihood on these values.

### 3.3. Landscape Architecture and Landscape Protection

The environment of vernacular baths is changed and shaped by nature every year. That is why it is difficult to build in a professionally acceptable way. Nature does not understand the installations of uninitiated, it destroys them and returns. The secret of building lies in understanding the place, which helps to create a built environment that co-exists organically with the place. The baths renovation programme provides a practical opportunity for young designers and students to learn and master traditional sustainable building methodologies, techniques and thinking. Through their ecological characteristics, design and the use of materials, the renovated baths will fit in with the character of the landscape and its local characteristics, thus enhancing its cultural value (Figure 11).

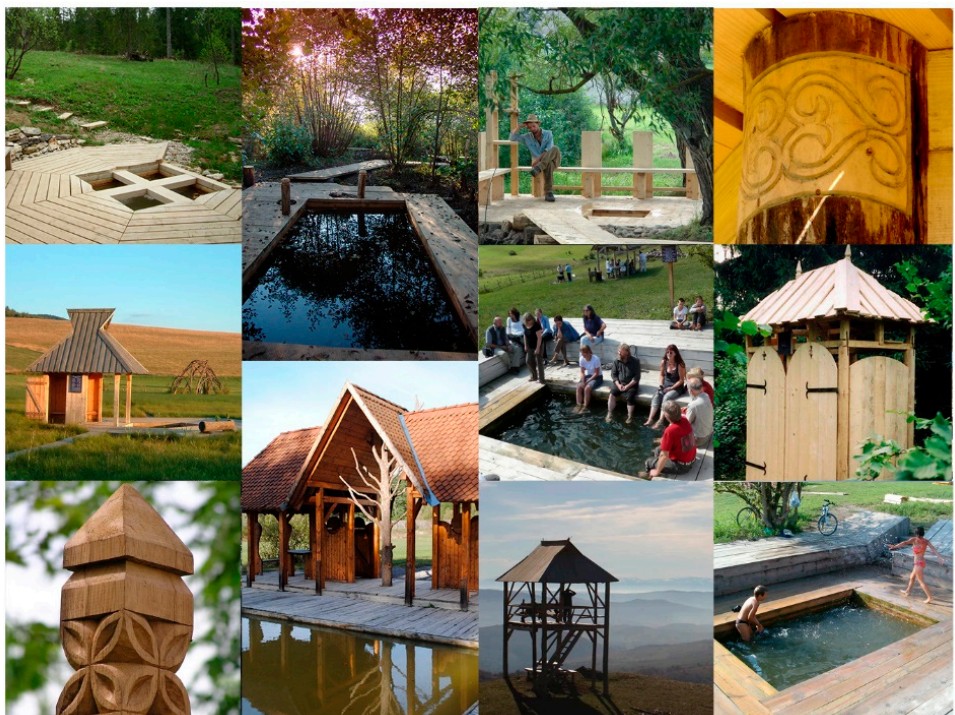

**Figure 11.** Detail photos of some renewed vernacular baths of community work. Source: Ars Topia Foundation.

The constructions are relatively simple: if necessary, a wooden spout is built upon the spring, which directs the water into a wooden basin sunk into a pit. The bottom and walls of the pool are insulated from the outside with clay. Sometimes a spring will fill a pit from the ground, without a visible wellhead. In this case, the mineral water flows in through the gaps in the wood. The beams of the bottom of the pool are laid on a gravel bed, and filtering layer is applied along the side walls to prevent the flowing water from washing the soil into the structure.

### 3.4. Community Building

Working together have been part of the life of the common man since ancient times. Tasks that were beyond the power of the individual could only be accomplished by working together. Instead of machines, relatives and neighbours helped to build a house, thresh the grain or pluck the feathers. Together, the work becomes easier, time is spent in conversation and fun, and the village community is built. People become to know each other, open up to the community, and are able to live their daily lives in community and harmony.

### 3.5. Keeping Traditions Alive

An important aim of the bath building volunteer camps is to keep traditions alive. It is not only about reviving the tradition of collaborative work, but also about practising

the traditional forms of construction. The materials used, the techniques employed, the forms and motifs used are all sourced from the world of simple-minded people, from local traditions. The built environment, based on the use of ancient, uniform use of materials and technology, also contributes to the characteristic visual appearance of the landscape specific to the location. The simple structures that support the use of the baths are among the last remaining examples of a type of architecture that is now dying out.

The ornamental motifs and architectural techniques (e.g., traditional carpentry) used in vernacular bath architecture in the second half of the 19th century are important and characteristic features of this environment [99,100].

These shapes and proportions still play a dominant role today, both in evoking the baths' heyday and in the nature of social and community life.

*3.6. Personal Development*

The camps have a great impact on all participants. In addition to shaping the environment, internal development is also important. People make friends, find role models and the evening activities help them to grow spiritually. From a professional point of view, landscape architects and architects have a significant experience of working with their hands and learning from local craftsmen. Everyone is shaped by the company, the atmosphere, the place and the work. The local village community by welcoming the participants as guests, and the volunteers by learning about local values and ways of thinking.

## 4. Conclusions

The strategy for the renovation of the vernacular baths in Szeklerland is a professional landscape architecture programme that has developed and gained recognition as the bottom-up initiative, based on social needs. The professional communities from Hungary, organised on the basis of the traditional knowledge of local communities, played a key role in the development of this strategy. The importance of these professional communities and the landscaping they initiated lies in the fact that, owing to their broad social recognition, the local or more distant partners who joined them have subsequently passed on the traditional landscape approach in many other places, in new professional partnerships.

A common feature of these professional communities is that they are strongly linked to the university-level landscape architecture education in Budapest. The university's clear mission is not only to educate but also to shape attitudes, and this must go beyond the classroom.

The Ars Topia Foundation is one of the pioneering professional workshops that set the direction of Transylvanian landscape architecture in the 21st century. The most significant result of the foundation's activities is undoubtedly the organisation of bath building volunteer camps in Szeklerland, the renewal of vernacular baths and local communities. The notion of landscape development based on heritage protection is also a key element in the activities of the Ars Topia Foundation. As intellectual and professional units, these professional communities have developed close cooperation with the landscape architecture higher education in Budapest (Institute of Landscape Architecture, Urban Planning and Garden Art, Hungarian University of Agriculture and Life Sciences, Budapest) and with the landscape architecture education in Transylvania (Sapientia EMTE University, Targu Mures, Romania). Following the camps organised, landscape surveys and surveys of valuable landscape features were also carried out with the participating landscape architecture students at several sites of the camps, organised by the Ars Topia Foundation. A general vision is to research and teach how development should be based on the values of the landscape, with respect to the traditional way of life of local societies. In recent years, the voluntary work of the professional communities has changed. The motivation of the participants has become more professional, and the projects have been increasingly formalised: they have been integrated into university courses, accepted as internship, and have been used and developed as topics for theses and dissertations, and even doctoral dissertations. It is no secret intention to support the professional development and networking of young

generations of landscape architects by organising joint field trips, design workshops and building camps, one of the best examples of which is the programme for the sustainable renewal and development of vernacular baths in Szeklerland, introduced in this article.

**Author Contributions:** Conceptualization, A.F., M.S. and Á.H.; methodology, A.F., M.S. and Á.H.; software, N.D.G.; validation, A.F., M.S. and Á.H.; formal analysis, N.D.G.; investigation, A.F., M.S. and Á.H.; resources, A.F., M.S. and Á.H.; data curation, Á.H.; writing—original draft preparation, A.F.; writing—review and editing, A.F. and M.S.; visualization, N.D.G.; project administration, A.F. and M.S. All authors have read and agreed to the published version of the manuscript.

**Funding:** This research received no external funding. The publication was supported by Hungarian University of Life Sciences—MATE.

**Institutional Review Board Statement:** Not applicable.

**Informed Consent Statement:** Not applicable.

**Data Availability Statement:** Not applicable.

**Acknowledgments:** We wish to thank the work of the anonymous reviewers.

**Conflicts of Interest:** The authors declare no conflict of interest.

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
