# Peer review of "Participatory Landscape Design and Water Management—A Sustainable Strategy for Renovation of Vernacular Baths and Landscape Protection in Szeklerland, Romania"

_land, doi:10.3390/land11010095_

Round 1

Reviewer 1 Report

This is a very interesting manuscript with many trans-disciplinary implications in landscape ecology, environmental planning, history, human dimension, and citizen management. Topic is very important and I read all with very interest. The ms is well written with a fluently language and rich of original points.  Figures are good and readable. Many photo are very original and useful for further projects in similar Camps.

I think that this ms deserves to be published on Land after only few MINOR REVISIONS. Here below I reported some minor points. However a good and original manuscript. I think that it will be readable by a large number of researchers of many different disciplinary arenas.

MINOR POINTS

Row 198. Add at least one reference.

Row 248. Delete comma after ‘literature’

Row 255. Perhaps in Italy (rich in mineral waters), some analogous sites are present?

Row 261 (‘traditional techniques’) and row 271-272: ‘with the involvement of the local community’ and: ‘realised with social participation’: Here and in other sections of the paper I suggest to add some papers about the process of Citizen Management (see Israel Journal of Ecology and Evolution, 2021, http://dx.doi.org/10.1163/22244662-bja10029). See also in the section ‘Community collaboration’ (row 304).

Row 278-279. ‘…building volunteer camps in the Carpathian Basin with a creative-educational purpose…’ Here I suggest to add a seminal reference: Jacobson, S. K., McDuff, M. D., & Monroe, M. C. (2015). Conservation education and outreach techniques. Oxford University Press. (also in the section ‘Environmental education’: row 335),

I am not a Mother Tongue. However, a further reading of English style and language could be necessary.

Add the role of the anonymous reviewers in the Acknowledgments.

Have good work.

Author Response

Responses to Reviewer No 1

We reworked the text integrating the specific remarks of the reviewers and we have followed the ratings of the criteria: reviewer 1, reviewer 2

We would like to thank the reviewers for their reviews, suggestions and remarks. It has greatly improved the content, form and text.

Point 1:

Row 198. Add at least one reference.

Three new references included.

Point 2:

Row 248. Delete comma after ‘literature’

Comma deleted.

Point 3:

Row 255. Perhaps in Italy (rich in mineral waters), some analogous sites are present?

According to our research, no similar sites in Italy (or Europe)

Point 4:

Row 261 (‘traditional techniques’) and row 271-272: ‘with the involvement of the local community’ and: ‘realised with social participation’: Here and in other sections of the paper I suggest to add some papers about the process of Citizen Management (see Israel Journal of Ecology and Evolution, 2021, http://dx.doi.org/10.1163/22244662-bja10029). See also in the section ‘Community collaboration’ (row 304).

Suggested paper added.

Point 5:

Row 278-279. ‘…building volunteer camps in the Carpathian Basin with a creative-educational purpose…’ Here I suggest to add a seminal reference: Jacobson, S. K., McDuff, M. D., & Monroe, M. C. (2015). Conservation education and outreach techniques. Oxford University Press. (also in the section ‘Environmental education’: row 335),

Suggested paper added.

Point 6:

I am not a Mother Tongue. However, a further reading of English style and language could be necessary.

Proofreading done.

Point 7:

Add the role of the anonymous reviewers in the Acknowledgments.

Added to the Acknowledgement.

Reviewer 2 Report

Review of land-1541123

This is an interesting paper, but it does not at present meet the standards of an academic publication. This can be readily addressed, however.

For ex, the text makes a number of unsubstantiated/unreferenced assertions. Such as

Line 111–122 needs referencing

Line 176–184 needs referencing

Line 218–221 needs referencing

Line 190          What storms of the 20th century? Give dates and references to these climatic extremes       or are you referring to political upheaval?

Figure 5.1 It would have been advantageous to see the site BEFORE intervention

Formal issues

Line 51            ‘mankind’       avoid gender excluding language. I this day and age such terms MUST not be used in academic writing.

Line 336          What is an ‘ unhidden aim’ ?

Figure 2           explain a–f please

Figure 3           the image is very messy and hard to read with all the underlining. Use different colours in the text

Figure 4           needs numbering and explanation

Please have the text corrected by a NATIVE English speaking professional editor. There are numerous unclear expressions as well as grammatical infelicities.

Author Response

Responses to Reviewer No 2

We reworked the text integrating the specific remarks of the reviewers and we have followed the ratings of the criteria: reviewer 1, reviewer 2

We would like to thank the reviewers for their reviews, suggestions and remarks. It has greatly improved the content, form and text.

The text makes a number ofunsubstantiated/unreferenced assertions. Such as:

Line 111–122 needs referencing

4 new references added

Line 176–184 needs referencing

4 new references and a new picture (Figure 5) added

Line 218–221 needs referencing

3 new references added

Line 190 - What storms of the 20 century? Give dates andreferences to these climatic extremes or are you referring to political upheaval?

We referred to the „unfavorable political developments” of that period. Corrected in the text.

Figure 5.1 It would have been advantageous to see the site BEFORE intervention

A new figure (Fig. 6) has been placed to show some ruined details and existing conditions before renewals

Line 51 - ‘mankind’ avoid gender excluding language. Ithis day and age such terms MUST not be used in academicwriting.

Changed into „humanity”

Line 336 What is an ‘unhidden aim’ ?

Changed into „One of the aims…”

Figure 2 explain a–f please

Explanation added

Figure 3 the image is very messy and hard to read with all the underlining. Use different colours in the text

We worked out several versions of the map, unfortunatelly no one better then this one. Due to the dense settlement network and the high amount of existing mineral water sources and local baths it is hard to show the right location of each site in charge. Moreover, the reduced available place on the page to fit the map contribute to this „messy” impression. Zooming in (which is easy to do as far as the resolution is good and the article will be published online) makes it more clear and understandable. 

Figure 4 needs numbering and explanation

Numerotation and explanation added

Please have the text corrected by a NATIVE English speaking professional editor. There are numerous unclear expressions as well as grammatical infelicities.

Proofreading done.